# Effect of an Electronic Medical Record-Based Screening System on a Rapid Response System: 8-Years’ Experience of a Single Center Cohort

**DOI:** 10.3390/jcm9020383

**Published:** 2020-02-01

**Authors:** Se Hee Lee, Chae-Man Lim, Younsuck Koh, Sang-Bum Hong, Jin Won Huh

**Affiliations:** Department of Pulmonary and Critical Care Medicine, Asan Medical Center, University of Ulsan College of Medicine, 88, Olympic-ro 43-gil, Songpa-gu, Seoul 05505, Korea; celestia7@gmail.com (S.H.L.); cmlim@amc.seoul.kr (C.-M.L.); yskoh@amc.seoul.kr (Y.K.); sbhong@amc.seoul.kr (S.-B.H.)

**Keywords:** clinical deterioration, early medical intervention, electronic health records, hospital rapid response team, intensive care units, medical records system, computerized

## Abstract

An electronic medical record (EMR)-based screening system has been developed as a trigger system for a rapid response team (RRT) that traditionally used direct calling. We compared event characteristics, intensive care unit (ICU) admission, and 28-day mortality following RRT activation of the two trigger systems. A total of 10,026 events were classified into four groups according to the activation time (i.e., daytime or on-call time) and the triggering type (i.e., calling or screening). Among surgical patients, the ICU admission was lowest for the on-call screening group (26.2%). Compared to the on-call screening group, the on-call calling group and daytime calling group showed higher ICU admission with an odds ratio (OR) of 2.07 (95% CI 1.50–2.84, *p* < 0.001) and OR of 2.68 (95% CI 1.91–3.77, *p* < 0.001), respectively. The 28-day mortality was lowest for the on-call screening group (8.7%). Compared to the on-call screening group, on-call calling (OR 1.88, 95% CI 1.20–2.95, *p* = 0.006) and daytime calling (OR 1.89, 95% CI 1.17–3.05, *p* < 0.001) showed higher 28-day mortality. The EMR-based screening system might be useful in detecting at-risk surgical patients, particularly during on-call time. The clinical usefulness of an EMR-based screening system can vary depending on patients’ characteristics.

## 1. Introduction

Rapid response teams (RRT) were widely deployed in the early 2000s to promptly detect deteriorating patients outside critical care and to provide appropriate advanced critical care early on [1]. RRTs are activated by calls by medical staff based on the calling criteria and the clinical concern. Increasing the RRT dose could improve patient outcomes [2,3]. However, previous research indicates that only 30% of at-risk patients who satisfied the calling criteria received critical care from RRTs [1]. In addition, diurnal variation affects RRT activation and clinical outcomes. RRT calls frequently occur during the day. Diurnal variation in RRT utilization influences hospital mortality dependent upon the time of the call [4,5]. Delayed RRT activation occurs more frequently between midnight and 8:00 am and is associated with increased hospital mortality [6]. Infrequent activation during early morning hours is followed by a spike in mortality at 7:00 am [7]. These findings suggest a delay in recognition of at-risk patients and suboptimal RRT utilization by caregivers at night results in poor patient outcomes.

Abundant clinical data and conclusions derived from electronic medical records (EMR) can be utilized to improve not only health care quality but also point-of-care management by detecting clinical deterioration early. The 24-h accessibility of EMR is beneficial in that automatic EMR monitoring by RRTs tends to reinforce the screening of at-risk patients. Currently, vital signs and certain laboratory data in EMR are used as criteria parameters for detecting deteriorating patients working as an additional limb of RRT [8,9]. However, results of EMR-based RRT systems have been mixed [9,10,11]. In our hospital, an RRT with dual-triggering afferent limbs, which utilizes both direct calling from bedside doctors or nurses and 24-h based EMR screening criteria, was introduced in 2008. We adopted a single-parameter EMR screening system. Previous research that assessed clinical outcomes in the first two-year period after dual triggering system deployment reported that EMR screening resulted in lower intensive care unit (ICU) admission rates but only surgically ill patients had reduced 28-day mortality rates [12]. This study aimed to analyze the event characteristics and clinical outcomes of RRT activations according to the trigger-type and activation time using the 8-year-period RRT cohort.

## 2. Methods

### 2.1. Study Populations

The study protocol was approved by the Institutional Review Board (2016-0857) of Asan Medical Center. Due to the retrospective nature, informed consent was not required, and patients’ data were used anonymously. This study was conducted at an academic tertiary care hospital with approximately 2400 adult beds. All adult patients in general wards who received treatment from the RRT were eligible. RRT operated for 24 h a day, 7 days a week during the study period. As the purpose of the study was to compare clinical characteristics and effectiveness of two triggering systems in early detection and management of at-risk patients, RRT events which were categorized as cardiopulmonary cerebral resuscitation (CPCR), post-CPCR care, educational purpose, procedure assistance, and counseling for end-of-life were excluded. Patients who requested to be listed as “do not resuscitate” (DNR) were also excluded. If the patient had more than one event in the same admission period, only the first event was included for analysis to eliminate redundancy.

Based on the duty hours of training residents, daytime for weekdays was defined as 7:00 am to 5:59 pm while on-call time was defined as 6:00 pm to 6:59 am on the following day. Daytime for weekends or holidays was defined as from 7:00 am to 11:59 am while on-call time was defined as from 12:00 pm to 6:59 am on the following day. Nursing staffs work in three shifts, day shift (6:30 am to 2:30 pm), evening shift (2:30 pm to 10:30 pm), and night shift (10:30 pm to 6:30 am on the following day), regardless of weekend or weekdays. We divided the type of activation events into four groups: calling vs. screening, based on trigger type, and daytime vs. on-call time, based on activation time.

At this hospital, the RRT not only provides advanced critical care but is also actively involved in monitoring and assessing at-risk patients throughout the day. Direct calling from ward physicians and nurses activates the RRT. Additionally, the EMR-based screening system is utilized, which automatically activates the RRT when the pre-defined criteria based on the vital signs and laboratory measurements of the patients’ medical records are met. Details of the criteria are shown in Appendix A.

### 2.2. Study Variables and Outcomes

Data were routinely collected for patients’ demographics, illness-type (medical or surgical), RRT activation time and date (weekdays or weekend), trigger parameter for activation, therapeutic intervention during the event (e.g., intubation, ventilator, high flow nasal cannular (HFNC), bilevel positive airway pressure (BiPAP), advanced cardiovascular life support (ACLS), etc.), the outcome of the RRT intervention (e.g., ICU transfer vs. ward stay), and the 28-day mortality following the event. If the alarm was triggered by both calling and screening, the first trigger was recorded. Patients’ vital signs (e.g., systolic/diastolic blood pressure (BP), pulse rate (PR), respiratory rate (RR), body temperature (BT), and mental status) at the time of the event were also collected to risk-stratify patients. We calculated a modified early warning score (MEWS), which was validated for both medical in-patients and surgical in-patients [13,14] and used this score for adjustment. MEWS is the sum of scores for five parameters: systolic BP, PR, RR, and mental status. Each parameter and detailed pre-assigned score are described in Appendix A.

The number of RRT events per year and the number of RRT activations per clock hour were analyzed to identify the RRT activation pattern. RRT events per each clock hour were classified as per screening and calling which were further divided into doctor-calling, and nurse-calling. All events were categorized into four different groups: daytime calling, daytime screening, on-call calling, and on-call screening. The primary outcome considered was ICU admission after RRT activation. Twenty-eight day mortality following RRT activation was also assessed for the four groups.

### 2.3. Statistical Methods

Annual RRT activations and the number of RRT events for each clock hour are presented graphically. Differences between two groups (i.e., calling vs. screening in daytime, and calling vs. screening in on-call time) were tested using a Chi-square test for categorical variables, and the independent t-test for continuous variables. ICU admission and 28-day mortality were compared between the on-call screening group and the other three individual groups and presented as an odds ratio (OR) with a 95% confidence interval using a multivariate logistic regression model adjusting for age, sex, MEWS, weekend, and activation coding after univariate analysis. The risk factors for ICU admission and 28-day mortality were identified following univariate logistic analysis and statistically significant variables were further applied for the multivariate logistic regression. All statistical analyses were performed using SPSS software (version 24.0; IBM Corp., Armonk, NY).

## 3. Results

From 1 January 2009 to 31 December 2016, 15,641 RRT events were identified (Figure 1); of these, 10,026 events were included for analysis. All events were classified according to activation time and trigger type. A total of 6293 events occurred during on-call time and 54.3% (*n* = 3419) were activated by screening rather than calling. Figure 2 represents the number of RRT activation events per year over the 8-year period. The number of RRT triggers by calling did not vary significantly between each year but activation by screening increased steadily from 2009. This increase was more prominent during on-call time.

The number of events for each clock hour are illustrated in Figure 3. In total, 4771 events (47.6%) were call-triggered (doctor calling and nurse calling). The number of activations by nurse calling was relatively stable in each clock hour compared with the number of doctor calling. RRT contacts were most frequent from midnight to 00:59 am (*n* = 952, 9.5%). The proportions of activation by screening were higher during on-call time, particularly from midnight to 0:59 at which time the vital check is conducted by night duty nurses.

Patients’ baseline characteristics, illness type, and MEWS are presented in Table 1. Approximately half of the patients in the screening group had solid malignancy (daytime 50.3% vs. 35.3%, *p* < 0.001; on-call time 50.2% vs. 41.5%, *p* < 0.001). Moreover, a greater number of patients had hematologic malignancy in the screening group than in the calling group (daytime 19.3% vs. 13.4%, *p* < 0.001; on-call time 16.7% vs. 11.6%, *p* < 0.001). In contrast, the proportion of patients with chronic lung disease, cardiovascular disease, or neurologic disease was higher in the calling group. Among surgical patients, a higher number of at-risk patients were activated by calling rather than screening (daytime 18.9% vs. 10.7%, *p* < 0.001: on-call time 19.6% vs. 12.9%, *p* < 0.001). MEWS was available in 9,736 events. MEWS was significantly higher in the calling group than in the screening group (daytime 4.54 vs. 4.30, *p* = 0.031; on-call 4.57 vs. 4.37, *p* = 0.0017). Activation coding and type of intervention are presented in Appendix A.

The overall ICU admission was 28.9% and 28-day mortality was 30% among 9736 patients. As more patients in the screening group had a malignancy (Table 1), a subgroup analysis was conducted to compare the clinical outcomes between patients with cancer and those without cancer (Table 2). Among patients with an underlying malignancy, the overall ICU admission rate was 21.5% and the on-call screening group displayed the lowest ICU admission (14.9%). The overall 28-day mortality was 40.6% and mortality was lower for the on-call calling group compared to the on-call screening group (OR 0.84, 95% CI 0.72–0.98). Among patients without cancer, overall ICU admission and 28-day mortality were 36.1% and 21.4%, respectively. ICU admission was lowest for the on-call screening group (22.3%) as was 28-day mortality (18.6%). Similar to patients with cancer, the daytime screening group had higher mortality compared to on-call screening group (OR 1.48, 95% CI 1.14–1.93). 

Among patients with surgical illnesses (Table 3) overall ICU admission and 28-day mortality were 37.6% and 13.4%, respectively. The on-call screening group was significantly associated with lower ICU admission (26.2%); daytime screening had an ICU admission of 28.4% (OR 1.06, 95% CI 0.70–1.59, *p* = 0.794), on-call calling 42.4% (OR 2.07, 95% CI 1.50–2.84, *p* < 0.001), and daytime calling 46.1% (OR 2.68, 95% CI 1.91–3.77, *p* < 0.001). The on-call screening group was also associated with lower 28-day mortality (8.7%); the 28-day mortality for daytime screening was 12.9% (OR 1.44, 95% CI 0.82–2.51, *p* = 0.203), on-call calling 15.5% (OR 1.88, 95% CI 1.20–2.95, *p* = 0.006), and daytime calling 15.9% (OR 1.89, 95% CI 1.17–3.05, *p* = 0.0009).

The risk factors associated with ICU transfer and 28-day mortality following RRT activation are shown in Appendix A. On-call time patients were less likely to be transferred to the ICU (OR 0.76, 95% CI: 0.68–0.84, *p* < 0.001) and had a lower mortality rate than daytime patients (OR 0.85, 95% CI: 0.78–0.94, *p* < 0.001). Compared to calling, the RRT activation by screening was associated with a lower ICU transfer rate, (OR 0.51, 95% CI: 0.46–0.57, *p* < 0.001) but a higher 28-day mortality rate (OR 1.19, 95% CI: 1.08–1.33).

## 4. Discussion

We evaluated the effects of an EMR-based screening system using the RRT cohort over an 8-year period following the employment of a dual triggering system. In addition to the calling system, the EMR-based screening system was implemented to aid in increasing the detection sensitivity of at-risk patients. Between 1 January 2009 and 31 December 2016, a total of 10,026 events were included. The number of RRT activations by screening system continually increased over the course of data collection. The higher proportion of screening group (especially on-call time) compared to the other study groups can be explained by two factors: (1) the experiences gained by the RRT over the study years might have led to an increase in the sensitivity of the screening system in detecting at-risk patients; (2) a small number of doctors on duty compared to daytime might result in a decrease in on-call calling, thus eventually increasing the burden of RRT work during on-call time.

Among the patients without cancer and surgical patients, the on-call screening group had lower ICU admission and lower 28-day mortality than other groups, possibly due to early detection using EMR screening during on-call time. However, this positive effect of the screening system was not observed among at-risk patients with cancer. Although more patients in the on-call calling group transferred to the ICU and had higher MEWS than the on-call screening group, 28-day mortality was significantly lower in the on-call calling group. The higher ICU admission rate in the calling group can be explained by the fact that activation by calling is (1) more likely to be associated with acute medical events and (2) more likely to reflect greater motivation on behalf of attending physicians to treat the patient. Alternatively, the daytime screening group had lower MEWS and lower ICU admission, but higher 28-day mortality. This result suggests that various factors affect mortality in medically ill at-risk patients. DNR agreement following RRT activation or consideration for end-of-life care might be closely related to 28-day mortality.

In many reports, the dose-response effect of the RRT was well described [15,16,17]. Increasing RRT dose was associated with dose-related reduction of cardiac arrest and cardiac arrests were most common overnight when RRT dose was the lowest. Because the trigger threshold by traditional calling criteria may vary depending on the experience or concern of ward physicians and nurses, the achievement of optimal RRT dose is important. Real-time monitoring by experienced RRT could increase the sensitivity of detecting clinical deterioration and improve clinical outcome. As our results indicate, triggering frequency itself is largely dependent on the interval of vital sign measurements. As vital sign check-ups at the ward are typically recorded by nurses at intervals of 8-h or 4-h, the activation frequency was higher during the regular vital sign check-up hours. Therefore, unless clinicians order frequent vital sign check-ups or laboratory tests for possible at-risk patients, the possibility of missing indicators of deteriorating patients will persist. Employment of automatic continuous monitoring could overcome this limitation [18].

We used single parameters including laboratory data, such as lactate levels and arterial blood gas analysis, as the triggers for the EMR-based system. Previous research indicates various degrees of sensitivity and the accuracy of multiple aggregate weighted scoring systems (AWSS) for predicting ICU transfer, cardiac arrest, and mortality [19,20]. However, the current AWSS depends mainly on vital signs, without assessing other characteristics of at-risk patients. Therefore, the development of a modified scoring system which includes vital signs, laboratory data, and characteristics of the patient population is necessary to increase the sensitivity and specificity of the screening system.

There are several limitations in this study. First, this is a single-center study and data were analyzed retrospectively. As our center is an academic tertiary hospital, the patients’ disease severity is generally higher than in non-tertiary hospitals; thus, our results may not generalize well to non-teaching hospitals or hospitals with different patient populations. Nevertheless, the use of a screening system during on-call time seems to be associated with an improved 28-day mortality rate, particularly among surgically ill patients and patients without malignancy. As our hospital adopted a dual-triggering system at the time of the launch of RRT, for ethical and patient safety concerns, a prospective randomized trial comparing calling system and screening system based on the duty hours was not possible. A prospective multicenter study is required to evaluate the efficacy of the screening system more accurately. Second, the proportion of patients who agreed on a plan for end-of-life care following RRT activation was not considered. There are other factors affecting primary outcomes aside from the management of RRT, such as end-of-life care, spontaneous decisions by attending physicians, will of family caregivers, and the availability of ICU beds at the time of the RRT visit. Therefore, these results should be interpreted with careful consideration of multiple clinical factors, not by the RRT intervention alone.

## 5. Conclusions

Deployment of an EMR-based screening system offers additional improvement in detecting and managing at-risk patients, particularly during on-call time. However, the clinical effectiveness of this system can vary depending on patients’ characteristics. The deployment of a modified screening system reflecting the physiologic parameters, laboratory measurements, and underlying diseases of the patient population at each hospital would maximize the beneficial role of the RRT in point of care management.

## Figures and Tables

**Figure 1 jcm-09-00383-f001:**
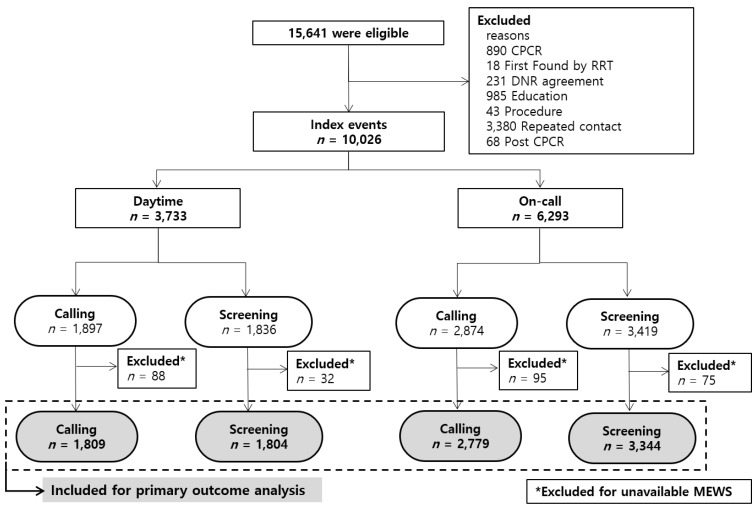
Schematic flow chart of the study Given that our rapid response team (RRT) performs a multifunctional role, only events related to early detection and management of at-risk patients were included for the study. Among 15,641 eligible events, 10,026 events were analyzed to describe the pattern of RRT activations. For clinical outcome analysis, 9736 events were included after excluding 290 events due to unavailable MEWS. Cardiopulmonary cerebral resuscitation (CPCR); Do not resuscitate (DNR); Rapid response team (RRT); Modified early weaning score (MEWS).

**Figure 2 jcm-09-00383-f002:**
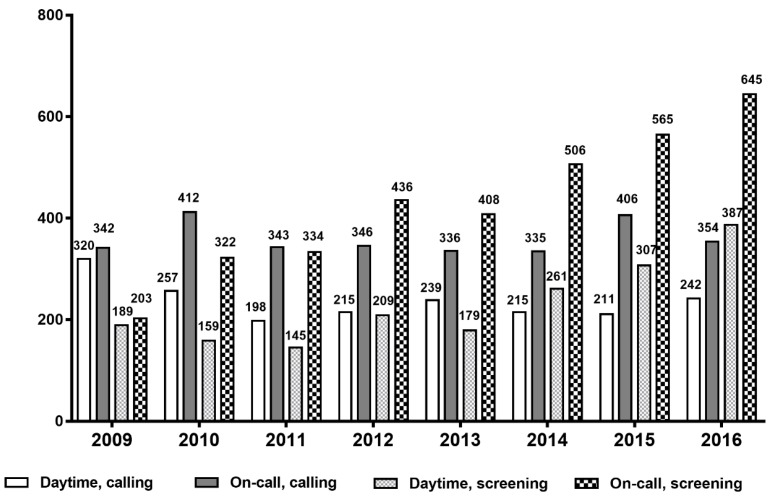
The number of RRT events per year since 2009. Overall RRT events increased from 1055 in 2009 to 1627 in 2016. The total number of RRT activations by screening in 2016 was 2.63-fold higher than that in 2009. Data are presented as number of events.

**Figure 3 jcm-09-00383-f003:**
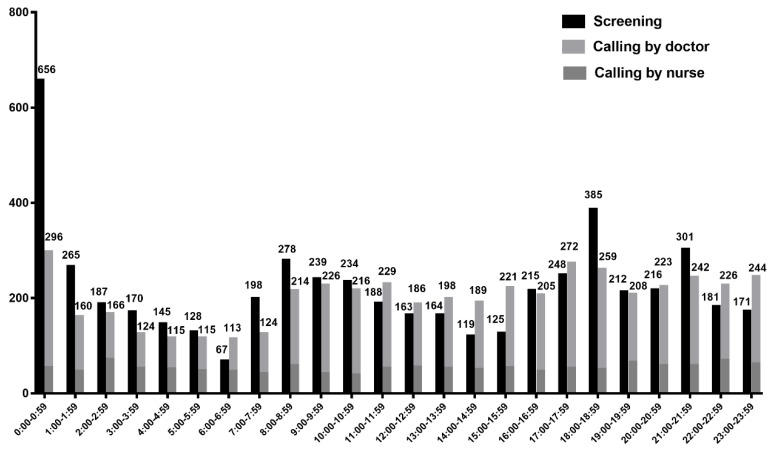
The RRT frequency according to each clock hour. Among 10,026 events, 4771 (47.6%) were triggered by calling and 5255 (52.4%) were triggered by screening. RRT contacts are most frequent at midnight to 00:59 am (*n* = 952, 9.5%). The total frequency was higher in order of 18:00 pm, 21:00 pm, and 8:00 am. Data are presented as number of events.

**Table 1 jcm-09-00383-t001:** Baseline characteristics of included events.

	Daytime		On-Call	
	Calling	Screening	Calling	Screening
	*N* = 1897	*N* = 1836	*N* = 2874	*N* = 3419
Age	64 (52–73)	64 (53–72)	64 (53–73)	64 (53–72)
Sex				
Male—No. (%)	1165 (61.7)	1120 (61.0)	1768 (61.5)	2097 (61.3)
Underlying disease				
Solid malignancy	669 (35.3)	924 (50.3) ^‡^	1194 (41.5)	1717 (50.2) ^‡^
Hematologic malignancy	254 (13.4)	355 (19.3) ^‡^	334 (11.6)	571 (16.7) ^‡^
Chronic lung disease	277 (14.6) *^*^*	224 (12.2)	405 (14.1) *^*^*	375 (11.0)
Cardiovascular disease	839 (44.2) ^†^	720 (39.2)	1327 (46.2) *^*^*	1483 (43.4)
Chronic liver disease	273 (14.4)	267 (14.5)	424 (14.8)	479 (14.0)
Gastrointestinal disease	7 (0.4)	6 (0.3)	16 (0.6)	14 (0.4)
Neurologic disease	262 (13.8) ^‡^	166 (9.0)	402 (14.0) ^‡^	322 (9.4)
Chronic kidney disease	158 (8.3) ^‡^	102 (5.6)	217 (7.6) *^*^*	203 (5.9)
Thyroid disease	95 (5.0) ^*^	61 (3.3)	113 (3.9)	140 (4.1)
Diabetes mellitus	440 (23.2)	429 (23.4)	703 (24.5)	825 (24.1)
Solid organ transplant	70 (3.7)	58 (3.2)	86 (3.0)	93 (2.7)
Illness type				
Medical	1450 (78.7)	1627 (88.6) ^‡^	2251 (79.9)	2987 (87.4) ^‡^
Surgical	392 (21.3) ^‡^	209 (11.4)	567 (20.1) ^‡^	432 (12.6)
MEWS	4.54 ± 2.23 ^‡^	4.30 ± 2.02	4.57 ± 2.24 ^‡^	4.37 ± 2.01
Weekend	330 (17.4)	377 (20.5) *^*^*	1103 (38.4)	1347 (40.2)

Among continuous variables, age is presented as median (interquartile range) and MEWS are presented as mean ± SD. Categorical variables are presented as No. (%). MEWS was available in 9736 patients. * *p*-value < 0.05, ^†^
*p*-value < 0.01, ^‡^
*p*-value < 0.001. Chi-square test was done for the comparison between daytime calling and daytime screening. The same analytic technique was used for the comparison between on-call calling and on-call screening. MEWS = modified early weaning score.

**Table 2 jcm-09-00383-t002:** Clinical outcomes among medical patients without cancer and with cancer.

		With Cancer (*N* = 4980)			Without Cancer (*N* = 3256)		
		OR	95% CI	*p*-Value	OR	95% CI	*p*-Value
ICU admission	On-call, screening	1			1		
	Daytime, screening	1.01	0.81–1.26	0.92	1.26	0.97–1.64	0.086
	On-call, calling	2.30	1.91–2.77	<0.001	2.42	1.98–2.97	<0.001
	Daytime, calling	3.85	3.11–4.77	<0.001	3.65	2.93–4.54	<0.001
28-day mortality	On-call, screening	1			1		
	Daytime, screening	1.16	0.99–1.35	0.063	1.48	1.14–1.93	0.004
	On-call, calling	0.84	0.72–0.98	0.026	1.10	0.88–1.38	0.417
	Daytime, calling	0.87	0.72–1.05	0.136	1.17	0.92–1.50	0.204

Data are presented as odds ratio (OR) with 95% confidence interval (CI). ICU admission and 28-day mortality were analyzed using a multivariate logistic regression model adjusting for age, sex, MEWS, weekend, and activation code. MEWS, weekend, and activation code were variables finally selected for the regression model for ICU admission in cancer patients. For 28-day mortality in cancer patients, sex, MEWS, weekend, and activation code were adopted variables for the regression model. Among patients without cancer, age, MEWS, and activation code were variables adopted for ICU admission and 28-day mortality. For patients with cancer: overall (*n* = 4980); on-call screening (*n* = 1971); daytime screening (*n* = 1131); on-call calling (*n* = 1188); daytime calling (*n* = 690). For patients without cancer: overall (*n* = 3256); On-call screening (*n* = 961); Daytime screening (*n* = 472); On-call calling (*n* = 1063); Daytime calling (*n* = 760).

**Table 3 jcm-09-00383-t003:** Clinical outcomes among patients with surgical illness.

		OR	95% CI	*p*-Value
ICU admission	On-call, screening	1		
	Daytime, screening	1.06	0.70–1.59	0.794
	On-call, calling	2.07	1.50–2.84	<0.0001
	Daytime, calling	2.68	1.91–3.77	<0.0001
28-day mortality	On-call, screening	1		
	Daytime, screening	1.44	0.82–2.51	0.203
	On-call, calling	1.88	1.20–2.95	0.006
	Daytime, calling	1.89	1.17–3.05	0.009

Data are presented as odds ratio with 95% confidence interval (CI). ICU admission and 28-day mortality were analyzed using a multivariate logistic regression model adjusting for age, sex, MEWS, weekend, and activation code. Sex, MEWS, weekend, and activation code were variables finally selected for the regression model for ICU admission. For 28-day mortality, sex, MEWS, and activation code were adopted variables for the regression model. Overall (*n* = 1500); on-call screening (*n* = 412); daytime screening (*n* = 201); on-call calling (*n* = 528); daytime calling (*n* = 359).

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
