# Peer review of "Effect of an Electronic Medical Record-Based Screening System on a Rapid Response System: 8-Years’ Experience of a Single Center Cohort"

_jcm, 2020, doi:10.3390/jcm9020383_

Round 1
Reviewer 1 Report
NoneAuthor Response
Thank you very much for considering the publication of our manuscript and for giving the opportunity to revise it.
Reviewer 2 Report
Thank you for the revisions you've implemented. A few additional comments to strengthen your work:
1. Abstract: Thank you for adding numbers to the abstract. Unfortunately I think you misunderstood which numbers to report and instead erred on reporting all numbers. The information you are providing is about the admission rates for each group. The corresponding question is 'Are these rates different' which can be answered by a simple chi-square test and a report of the percentages (as you have) and with p-value at the end of the comparison sentence. If you want to report Odds Ratios, that's also fine -- but it answers a different question. It says these are the odds of admission in group X compared to group Y. Choose which you are reporting, or both, and update the text accordingly.
2. In section 2.1, paragraph 2: I apologize. I'm still confused by the way daytime and on-call time was defined. Are the times provided only applied to residents / doctors? How was this defined for nursing staff since both of these staff types can activate an event, and nursing staff have three shifts?
2. Section 2.3, Thank you for the improved description. Again, the statistical analysis for every table needs to be accurately described. Table 1 is not a comparison across all four groups. It is two analyses in one table, e.g. calling vs. screening in daytime, and calling vs. screening in on call time. Why is the Mann-Whitney test (which is non-parametric) being used? This is a non-parametric test. Table 2 is a stratified analysis based on cancer status. Table 3 is a sub-set analysis of patients with surgical status. If you had a process to select which variables were included in the multivariate analysis, indicate what that was. A p-value requirement? AIC? What variables were in the potential confounder pool. If it's all those listed in Table 1, you could reference this as the list and simply write something similar to: "age, sex, disease type, illness type, MEWS as listed in Table 1 were considered."
Also great job on improving the readability and clarity of the tables and figures! This made your story so much easier to understand. Nice work!
Author Response
Reviewer reports:
Reviewer #2:
Abstract: Thank you for adding numbers to the abstract. Unfortunately I think you misunderstood which numbers to report and instead erred on reporting all numbers. The information you are providing is about the admission rates for each group. The corresponding question is 'Are these rates different' which can be answered by a simple chi-square test and a report of the percentages (as you have) and with p-value at the end of the comparison sentence. If you want to report Odds Ratios, that's also fine -- but it answers a different question. It says these are the odds of admission in group X compared to group Y. Choose which you are reporting, or both, and update the text accordingly.a. We assume the binary event outcome, ICU admission and 28-day mortality after RRT activation, would be compared by logistic regression analysis. The reported p-value and odds ratio was the result of multivariate logistic regression analysis. We apologize to you for causing confusions and we presented the numbers with odds ratio.
Revised manuscript abstract
Among surgical patients, the ICU admission was lowest for the on-call screening group (26.2%). Compared to the on-call screening group, the on-call calling group and daytime calling group showed higher ICU admission with an odds ratio (OR) of 2.07 (95% CI 1.50-2.84, p < 0.001) and OR of 2.68 (95% CI 1.91-3.77, p < 0.001), respectively. The 28-day mortality was lowest for the on-call screening group (8.7%). Compared to the on-call screening group, on-call calling (OR 1.88, 95% CI 1.20-2.95, p = 0.006), and daytime calling (OR 1.89, 95% CI 1.17-3.05 p < 0.001) showed higher 28-day mortality.
In section 2.1, paragraph 2: I apologize. I'm still confused by the way daytime and on-call time was defined. Are the times provided only applied to residents / doctors? How was this defined for nursing staff since both of these staff types can activate an event, and nursing staff have three shifts?
a. I apologize for the unclarity of the sentence. Regular duty hours for doctors, including attendings, residents, and fellows, from Monday to Friday is 7:00 AM to 6:00 PM. On weekends and holidays, duty hours for residents and fellows is 7:00 AM to 12:00 PM. On-call time is defined as time other than regular duty hours. Nursing staffs work on a three-shift cycle; day shift (6:30 AM to 2:30 PM), evening shift (2:30 PM to 10:30 PM), and night shift (10:30 PM to 6:30 AM on the following day).
Revised manuscript methods page 2 line 67-68
Nursing staffs work in three shifts, day shift (6:30 am to 2:30 pm), evening shift (2:30 pm to 10:30 pm), and night shift (10:30 pm to 6:30 am on the following day), regardless of weekend or weekdays.
Section 2.3, Thank you for the improved description. Again, the statistical analysis for every table needs to be accurately described. Table 1 is not a comparison across all four groups. It is two analyses in one table, e.g. calling vs. screening in daytime, and calling vs. screening in on call time. Why is the Mann-Whitney test (which is non-parametric) being used? This is a non-parametric test. Table 2 is a stratified analysis based on cancer status. Table 3 is a sub-set analysis of patients with surgical status. If you had a process to select which variables were included in the multivariate analysis, indicate what that was. A p-value requirement? AIC? What variables were in the potential confounder pool. If it's all those listed in Table 1, you could reference this as the list and simply write something similar to: "age, sex, disease type, illness type, MEWS as listed in Table 1 were considered."
a. Thank you for your comment. We initially analyzed baseline characteristics across all four groups, and then analyzed between two groups, i.e. calling vs. screening in daytime, and calling vs. screening in on-call time for the simplicity. As you comment, we modified the sentence. We apologize for the error in statistical analysis. Considering sample numbers in each group, parametric test is an appropriate test for continuous variables (e.g, age and MEWS). An independent t-test is a correct test. We modified the related sentence, and test result accordingly.
Revised manuscript methods page 3 line 96-98
Differences between two groups (i.e., calling vs. screening in daytime, and calling vs. screening in on-call time) were tested using a Chi-square test for categorical variables, and the independent t-test for continuous variables.
b. We selected variables among different clinical parameters (age, sex, weekend, MEWS, underlying disease, trigger type, and activation code) by using a univariate regression analysis with cut-off p-value 0.05. After selecting the covariates, we underwent a multivariate regression analysis with a backward method (LR). We used the Hosmer-Lemeshow test to verify the fitness of the model instead of AIC.
Revised manuscript methods page 8 Table 2 line 180-183
Data are presented as odds ratio (OR) with 95% confidence interval (CI). ICU admission and 28-day mortality were analyzed using a multivariate logistic regression model adjusting for age, sex, MEWS, weekend, and activation code. MEWS, weekend, and activation code were variables finally selected for the regression model for ICU admission in cancer patients. For 28-day mortality in cancer patients, sex, MEWS, weekend and activation code were adopted variables for the regression model. Among patients without cancer, age, MEWS, activation code were variables adopted for ICU admission and 28-day mortality.
Revised manuscript methods page 9 Table 3 line 180-183
ICU admission and 28-day mortality were analyzed using a multivariate logistic regression model adjusting for age, sex, MEWS, weekend, and activation code. Sex, MEWS, weekend, and activation code were variables finally selected for the regression model for ICU admission. For 28-day mortality, age, sex, MEWS, activation code, and weekend were adopted variables for the regression model.
This manuscript is a resubmission of an earlier submission. The following is a list of the peer review reports and author responses from that submission.
Round 1
Reviewer 1 Report
Lee and coll. describe the effects of an EMR based screening system using an 8-year cohort of rapid response team interventions. Results suggested a posive effect of this screening system in surgical and non oncologic patients.
This result is not particularly unexpected but represents an interesting starting point for future studies.
I suggest the following minor revisions.
1) A significant percentage of patients (5615) were excluded from the analysis.
The reason for the exclusion should be better explained. More precisely, the exclusion criteria should be clearly described in the methods section. Furthermore, the legend in Figure 1 should be more explanatory in this regard.
2) There is no correspondence between the total number of eligible patients indicated in the text (page 3, line 91) and that reported in figure 1.
Reviewer 2 Report
Thank you for the opportunity to review your work. This manuscript discusses the use of two RRT triggers; 1) via an EMR-based screening system and 2) traditional direct calling. Events were compared based on day-time and on-call timeframes and considered ICU admission rates and 28-day mortality as the outcomes.
Overall:
This manuscript is filled with grammatical errors. For example, in many case the word ‘the’ is often missing. I had a hard time reading and comprehending the content, because this was so prolific.
Abstract:
The abstract fluctuates between full sentences and shortened snippets. There are a lot of small words, such as the and a, as well as verbs that are missing.
Admission rates are reported but there is no indication if these are significant differences, and if so, what that significance level is. There are no confidence intervals or p-values.
The sub-group analysis is indicated as being significant, but no confidence limits / p-values are reported. Statistically significant doesn't necessarily mean clinically meaningful.
Introduction:
It was confusing to see the common abbreviation of EMR not used as the common definition; e.g. EMR = Electronic Medical Record. I would adjust this throughout the manuscript for clarity and readability.
The word limb in not used correctly.
Methods:
It didn’t make sense to me that on-call time did not include weekends. The hospital systems and most departments I’m familiar with work with a reduced staff on Saturdays and Sundays. If this wasn’t the case in this setting, that should be described.
How was illness type defined?
Please provide more information about the modified early warning score. There is no reference to indicate that this is a validated score. If using it for illness severity adjustment as well, the reader should know all components that make up this score and how this score was calculated.
The primary outcome was ICU admission rate. All in-hospital patients were included, which would include those already in the ICU. What was used if the patient was already in the ICU? Were patients admitted to the ICU excluded?
How were the annual changes in RRT activation calculated?
How was the 28-day in-hospital mortality rate calculated, if patients were discharged? This phrasing is confusing if it includes patients who were discharged within this time frame. Or potentially what is meant here is 28-day mortality from RRT intitiation?
Is this a patient level analysis with the outcome being a binary yes-no for patient ICU admission? If yes, the primary outcome isn’t really ICU admission rates. It is ICU admission. Please specify the model structure used to produce the odds ratios and confidence intervals. Were univariate models also calculated? If yes, this should be described. The analysis for every table provided or discussed in the results should be described, even if it is to say the ‘same analytical technique was used to….’
Was there any consideration if those excluded due to a missing MEWS were different from those who were included?
Results:
What is CPCR? This acronym is not defined and yet used in Figure 1.
I don’t see how Figure 2 illustrates the annual changes in RRT events. It instead is showing the count of RRT events per year based on the four sub-groups created by the study team.
It isn’t clear that Figure 3 is the total of all RRT events broken down by hour. I believe the numbers reported are not cumulative, but I’m not certain as this is not explained in the methods section.
Figure 3: The numbers are difficult to read.
Table 1: Which variables are reported as mean (SD) and which as median (Interquartile Range)?
Table 1: Are all significant p-values greater than p=0.01? Why is there not a three-level p-value symbol system consisting of p<0.05, p<0.01, and p<0.001? How else do you describe the categorical variables, if not as N(%)? The same symbol is used in the variable list as to describe significance. These should be different symbols.
Table 2: A single number does not equate to a confidence interval. This is not described appropriately. It appears that multiple tests are being reported in this table, but it isn’t clear what they are. I don’t understand why p-values are given in the last column and yet a p-value symbol being used in the N(%) or score column.
Table 3: Has many of the same issues as Table 2.
Tables should be self-sufficient. When one looks at a table, they should have all the necessary information to understand not only what analysis was used, but to clearly understand the results. Neither Table 2 or Table 3 fit this criterion and should be re-worked for clarity.
I briefly looked at the supplementary tables. I suggest the same types of re-working to make these more easily understood by a reader. In my opinion, there is too much information contained in this manuscript. There are four groups and the comparisons being made are two groups at a time, but this isn’t totally clear in the methods which means it also not fully clear in the results or tables.
I also don’t understand why in some places all four groups are compared (with on call, screening as the reference) while in other analyses this is done in pairs. I’d think it would make sense to have two variables, one for on-call vs. daytime and another for screening vs. calling so you can determine the effect of each component separately with potentially an interaction term?
At the same time, a large portion of the results and the discussion seems to reference supplementary material.
Discussion
Couldn’t an illness severity measure such as APACHE be used, if one has access to vitals, labs, and arterial blood gases? What additional patient characteristics would be suggested?
It is mentioned that ICU bed availability is a limiting factor. Was this something the study team considered or data that was accessible?